

# Multivariate ordination identifies vegetation types associated with spider conservation in brassica crops

Hafiz Sohaib Ahmed Saqib[1,2], Minsheng You[1,2,3] and Geoff M. Gurr[1,2,3,4]

[1] State Key Laboratory of Ecological Pest Control for Fujian and Taiwan Crops, Fujian Agriculture and Forestry University, Fuzhou, China
[2] Institute of Applied Ecology, Fujian Agriculture and Forestry University, Fuzhou, China
[3] Fujian-Taiwan Joint Centre for Ecological Control of Crop Pests, Fujian Agriculture and Forestry University, Fuzhou, Fujian, China
[4] Graham Centre for Agricultural Innovation, Charles Sturt University, Orange, New South Wales, Australia

## ABSTRACT

Conservation biological control emphasizes natural and other non-crop vegetation as a source of natural enemies to focal crops. There is an unmet need for better methods to identify the types of vegetation that are optimal to support specific natural enemies that may colonize the crops. Here we explore the commonality of the spider assemblage—considering abundance and diversity ($H$)—in brassica crops with that of adjacent non-crop and non-brassica crop vegetation. We employ spatial-based multivariate ordination approaches, hierarchical clustering and spatial eigenvector analysis. The small-scale mixed cropping and high disturbance frequency of southern Chinese vegetation farming offered a setting to test the role of alternate vegetation for spider conservation. Our findings indicate that spider families differ markedly in occurrence with respect to vegetation type. Grassy field margins, non-crop vegetation, taro and sweetpotato harbour spider morphospecies and functional groups that are also present in brassica crops. In contrast, pumpkin and litchi contain spiders not found in brassicas, and so may have little benefit for conservation biological control services for brassicas. Our findings also illustrate the utility of advanced statistical approaches for identifying spatial relationships between natural enemies and the land uses most likely to offer alternative habitats for conservation biological control efforts that generates testable hypotheses for future studies.

## INTRODUCTION

In recent decades, anthropogenic activities—such as land clearing, environmental pollution and agricultural intensification—have led to adverse effects on the occurrence, diversity and evenness (*Bengtsson, Ahnström & Weibull, 2005*; *Benton, Vickery & Wilson, 2003*; *Landis, Wratten & Gurr, 2000*; *Sunderland & Samu, 2000*; *Thies et al., 2011*; *Thies & Tscharntke, 1999*), and even the outright extinction of numerous species (*Thomas et al., 2004*). Biodiversity loss due to agricultural intensification is not merely driven by increases

Corresponding author
Geoff M. Gurr, ggurr@csu.edu.au

in the non-judicious use of hazardous fertilizers and pesticides (*Geiger et al., 2010*; *Roubos, Rodriguez-Saona & Isaacs, 2014*), but also the landscape simplification and fragmentation, and the loss of habitat on which many species rely. To limit the use of chemical inputs and to fulfill the food demands of a growing worldwide population, researchers and growers have shifted their attention to the development of effective integrated pest management (IPM) tactics by manipulating the cultural farming practices including vegetation patterns (*Gurr et al., 2016*; *Gurr et al., 2017*; *Landis, Wratten & Gurr, 2000*), often specifically to conserve biological control agents (*Fiedler, Landis & Wratten, 2008*; *Liu et al., 2014*; *Pedigo & Rice, 2014*).

Habitat management has long been used to promote beneficial arthropods in agroecosystems for the delivery of ecosystem services, particularly biological pest control (*Gurr et al., 2017*). The addition of non-crop vegetation to a crop system is effective in enhancing local densities of predators and parasitoids but is often not readily compatible with farming practices and may reduce yields by reducing the area sown to the crop (*Letourneau et al., 2011*). An alternative approach is to manipulate the availability of nearby donor habitat in field margins or adjacent fields and uncropped zones. This avoids the need to reduce to the extent of the focal crop. There is a need, however, to develop approaches that will help understand specific interactions between crops, adjacent vegetation types and natural enemies (predators and parasitoids) (*Furlong, 2015*; *Furlong et al., 2008*; *Furlong & Zalucki, 2010*; *Szendrei et al., 2014*; *Tscharntke et al., 2012*).

Addressing the foregoing challenge has been methodologically difficult but population and community ecology have entered an exciting phase of pattern unification (*Blanchet, Legendre & Borcard, 2008*; *Legendre & Gauthier, 2014*). As the importance of spatial ecological models has become better understood (*Legendre & Fortin, 1989*; *Legendre & Gauthier, 2014*), it has become increasingly clear that ecologists need to incorporate these spatial distribution patterns into their ecological models. There have been a number of methodological developments in ecology to investigate the influence of environmental gradients on species' spatial distribution patterns (*Legendre & Gauthier, 2014*), for example incorporation of geostatistical tools to explain geographical variation of species (*Peterson, Theobald & Hoef, 2007*). Spatial autocorrelation analysis is more robust and forgiving of lower sample sizes and missing data that often accompany agroecological studies, compared with the classical geostatistical approaches (e.g., semivariograms) (*Blanchet, Legendre & Borcard, 2008*; *Legendre & Gauthier, 2014*). There are several reasons to measure spatial autocorrelation in studies of this nature. First, it indexes the nature and extent to which fundamental statistical assumptions are violated, and, in turn, indicates the degree to which conventional statistical inferences are compromised. It also signifies the presence of and quantifies the extent of redundant information in georeferenced data, which in turn affects the information contribution of each georeferenced observation to statistics calculated with a database. More fundamentally, the measurement of spatial autocorrelation describes the overall patterns across a geographic landscape, supporting spatial prediction and allowing detection of striking deviations (*Griffith, 2013*).

Spiders (Araneae) are an invariably abundant and dominant, species-rich guild of predators in crop fields (*Marc, Canard & Ysnel, 1999*; *Nyffeler & Sunderland, 2003*; *Schmidt et al., 2003*; *Schmidt & Tscharntke, 2005*). Characteristically, few spider taxa achieve dominance on agricultural lands, and they have been referred to as "agrobionts" (*Marc, Canard & Ysnel, 1999*; *Samu & Szinetár, 2002*) and can play a vital role not only as generalist predators in suppressing the pest densities, but also as specialist predators of key pest species. For example, *Chapman et al. (2013)* showed that spider species are not truly polyphagous, but exhibit the specialized feeding habits by feeding on jumping prey items such as Collembola or slowly-crawling prey such as aphids. The results of another study also suggested that manipulating spider community composition to give complementary functional groups (i.e., foliage-hunters *Xysticus cristatus* (Thomisidae) and the ground-hunters *Pardosa palustris* (Lycosidae)), can give a better biological control compared with conserving predator biodiversity per se which can occur without necessarily increasing functional diversity (*Birkhofer et al., 2008*). Earlier work, *Riechert & Lawrence (1997)* and *Riechert & Bishop (1990)*, showed that the significant effect of spiders on the suite of pests in a mixed vegetable cropping systems was an assemblage effect, rather than the effect of just a few dominant spider species. It can, therefore, be important to focus conservation biological control efforts relatively broadly across multiple natural enemy functional groups.

It is, however, not clear if spider species utilise agricultural habitat in general or exhibit specificity to crop and non-crop habitats on farms. This has clear and important ramifications for the extent to which spiders utilize a diversity of crop types and non-crop vegetation as source habitat when colonizing a focal crop of interest. This study was designed to explore the extent of the similarity between spider assemblages in brassica crops and different types of adjacent (non-brassica) crop and non-crop vegetation, and to explore the influence of various adjacent vegetation types on the spatial distribution of spiders. Specifically, we hypothesized that abundance and diversity—including functional groups—of spiders would differ among vegetation types represented in a brassica-production landscape, that some vegetation types would have spider assemblages similar to that of brassica crops, and that this would indicate the potential value of this vegetation as donor habitat from which spiders could move to colonise a newly planted brassica crop or to repopulate after a disturbance event.

## MATERIALS AND METHODS

### Experimental design and sampling

Spiders were sampled in brassica crops and adjacent vegetation types in three sites in Fujian Province, China. Two sites were located in the Nantong district (25°55′13.97″N, 119°15′42.15″E & 25°55′0.25″N, 119°15′39.46″E, respectively) and a third in the Minqing district (26°10′4.72″N-118°46′18.08″E), of greater Fuzhou City. Each site comprised a focal brassica field and the adjacent vegetation types (comprised of both crop and non-crop habitats) within an approximate 50 × 50 m grid. Adjacent crop habitats included litchi, pumpkin, sweetpotato and taro; whilst non-crop habitat types consisted of adjacent field

margins and fallow fields (both containing a variety of grasses, forbs and some bare ground) as well as non-crop vegetation with small woody perennials. The three sites were typical of smallholder farming in southeastern China and common in other agricultural systems globally. All of the agronomic practices—including fertilizer inputs and (frequent) pesticide application—were carried out as per normal by the host farmers.

At each site, spiders were sampled from at least 25 and up to 29 grid points (Minqing $n = 29$ points, Nantong 1 $n = 25$ points, Nantong 2 $n = 27$ points) (at least 10 m apart) extending across adjacent vegetation types to the brassica field. Samples were collected on five occasions from August and December of 2015, using a motorized blower-vacuum sampler (YAHAMA-EBV260) with a removable net bag mounted in the inlet (*Lee et al., 2014*; *Lin, Vasseur & You, 2016*; *Whitehouse, Wilson & Fitt, 2005*). A major typhoon in October completely flooded fields, which severely affected the population dynamics of spiders. Two sampling events before typhoon were considered for analysis, while three sampling occasions collected after the typhoon were not considered in the analysis, as spider abundances were very low. Samples were collected at each grid point by running the vacuum sampler for 2 min within an area of 2 m$^2$. Sample bags were labeled and transferred to an ice box to prevent predation and sample degradation, and taken to the laboratory for sorting and identification under a stereo microscope. All of the samples were kept in 95% ethanol (EtOH) for preservation. Adults and immatures were identified to family level and assigned to the morphospecies using BOLD taxonomic classifications (*Ratnasingham & Hebert, 2007*) and a morphological key (*Carl, 2016*). Global Positioning System (GPS) data of xy-coordinates were recorded using GARMIN GPS device (GPSMAP$^{\circledR}$ 60CSx).

## Statistical analysis

To test the importance of vegetation types on spider assemblages in brassica fields and the influence of those habitats on the spatial distribution of spider species, we applied variance partitioning, hierarchical clustering (for community similarities or dissimilarities) and spatial eigenvector analysis for spider abundance and diversity data. Abundance "$n$" and Shannon-Wiener index "$H$" (*Shannon et al., 1949*) were calculated using the vegan package (vegan 2.4-0) (*Oksanen et al., 2016*), in R statistical software (R version 3.4.0), then the data were Hellinger transformed to obtain normality and adjust variance prior to multivariate analysis. The Hellinger transformation has good statistical properties to test for relationships among explanatory variables and draw biplots in constrained or unconstrained multivariate ordination (e.g., redundancy analysis RDA) without resorting to the Euclidean distances (*Legendre & Gallagher, 2001*) and is also suited to data sets with multiple zero values. We identified the response of spider abundance and diversity ($H$) against different vegetation types and weighted principal coordinates of neighbor matrices (PCNM) as explanatory variables using the "varpart" and "pcnm" functions of package "vegan" (version 2.4-1) (*Oksanen et al., 2016*) in R (version 3.4.0), which allowed variance partitioning to separate the effects of weighted PCNM and vegetation types on spider abundance and diversity ($H$) (*Peres-Neto et al., 2006*). PCNM, also known as Moran's Eigenvector Maps (MEM), is a powerful approach able to detect spatial or temporal patterns (henceforth, only spatial patterns will be discussed) of varying scale in response

data (spider abundance and diversity) (*Borcard & Legendre, 2002*; *Borcard et al., 2004*; *Dray, Legendre & Peres-Neto, 2006*). Essentially, spatial variables are used to determine the distance between sites with special focus on neighbouring sites. Additionally, the "rda" function of package "vegan" (version 2.4-1) was used to test the significance of fractions of each spider family's abundance and diversity ($H$), and triplots were constructed to visualize the vegetation types associated with different spider families. All analyses was carried out separately for each of the three experimental sites because of differences in adjacent vegetation types to the brassica field.

To measure community dissimilarities of spiders in different vegetation types, hierarchical clustering was carried out for the abundance and diversity ($H$) per sampling points at each experimental site. A quantitative version of the Sørensen index, Bray-Curtis dissimilarity was used to measure the percentage differences and to construct dissimilarity matrices for abundance and diversity ($H$) of spider families in brassica and adjacent crop and non-crop habitat types using the "vegdist" function with "method = "bray"" (*Aanderud et al., 2015*; *Jeremy, 2013*) using "vegan" (version 2.4-1) (*Oksanen et al., 2016*). We visualized the $\beta$-dissimilarity matrix using heatmap for the abundance and diversity ($H$) of spider families at each of the experimental sites (*Aanderud et al., 2015*; *Jeremy, 2013*; *Murtagh & Legendre, 2014*) by using the "gplots" (*Gregory, Warnes & Lodewijk, 2016*), "Heatplus" (*Ploner, 2015*), "RColorBrewer" (*Neuwirth, 2014*) and "ComplexHeatmap" (*Gu, Eils & Schlesner, 2016*) packages in R (version 3.4.0). An assessment of the uncertainty in the cluster delineation was done through multiscale nonparametric bootstrap resampling tests (*Shimodaira, 2002*) using "pvclust" (*Suzuki & Shimodaira, 2013*) package in R (version 3.4.0). This helps to determine $p$-values (two types: approximately unbiased (AU) $p$-value and bootstrap probability (BP) value) of each cluster in the hierarchy (*Suzuki & Shimodaira, 2006*).

Spatial eigenvector analysis is particularly well suited to data with low spatial or temporal replication, when compared to classical geostatistical analysis (e.g., semivariograms) (*Peres-Neto & Legendre, 2010*; *Perović & Gurr, 2012*), which was the case in our data. We were interested in calculating and mapping the spatial variation in the occurrence of spiders, and analyzing its relationship with the adjacent vegetation of the focal brassica field. Distance-based MEM (dbMEM) (*Borcard et al., 2004*; *Legendre & Gauthier, 2014*) was used to control for spatial autocorrelation in tests of abundance and diversity ($H$) of spider-vegetation relationships, see *Griffith & Peres-Neto (2006)* using the packages "adespatial" (*Stéphane et al., 2017*), "ade4" (*Chessel, Dufour & Dray, 2009*), "adegraphics" (*Stéphane & Aurélie, 2017*) in R (version 3.4.0). We identified a total of 11 distance based Moran's eigenvector maps for Minqing, seven for Nantong 1 and nine for Nantong 2. Significant Moran's eigenvector maps for each of the experimental sites were identified with forward selection using double stop criterion (*Blanchet, Legendre & Borcard, 2008*), $\alpha = 0.05$ and $R^2$ values (for abundance; $R^2 = 0.45$ in Minqing, $R^2 = 0.37$ in Nantong 1 and $R^2 = 0.34$ in Nantong 2, and for diversity ($H$); $R^2 = 0.46$ in Minqing, $R^2 = 0.34$ in Nantong 1 and $R^2 = 0.23$ in Nantong 2). We identified one significant Moran's eigenvector map for spider abundance out of a total of 11 in Minqing and nine for Nantong 2. Whilst for diversity ($H$); we identified two significant Moran's eigenvector maps out of a total of

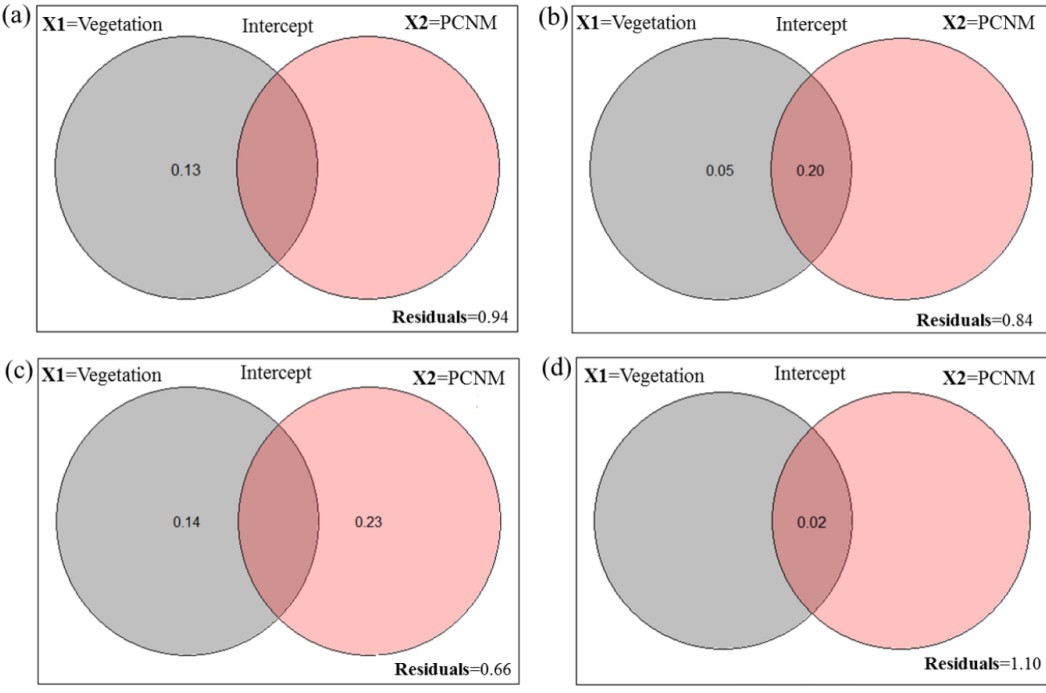

**Figure 1** Venn diagram for the fractions of variation obtained by variance partitioning of a response data set; "Y" = Hellinger transformed spider taxa (A) abundance at Minqing, (B) diversity ($H$) at Minqing, (C) diversity ($H$) at Nantong 1 and (D) diversity ($H$) at Nantong 2; against two explanatory environmental variable matrices; "X1" = Vegetation type surrounding the brassica field and "X2" = Principle Coordinates of Neighborhood Matrix (PCNM) and their intercept.

11 in Minqing and one out of nine for Nantong 2. Further, canonical analysis (rda) was performed to compute the dbMEM spatial models and the "anova" function was used to test the significance of these models. All spatial models were found to be highly significant ($p$-value < 0.001). R-codes and datasets are attached as Data S1–S7.

## RESULTS

A total of 919 (461-Minqing, 216-Nantong 1 and 242 at Nantong 2) individual spiders were captured, representing 48 morphospecies across nine families. In Minqing, variance partitioning results showed that vegetation type (X1) alone explained 13% of variation in abundance of spiders, and the total effect of X1 and PCNM (X2) was 6% (Fig. 1A). On the other hand, 5% of variation in diversity ($H$) of spiders at Minqing alone was explained by the variable X1, and 20% of variation was explained by the X1 + X2 (intercept), whilst the total effect of both variables X1 and X2 was 16% (Fig. 1B). The 23% of variation in spider diversity at Nantong 1 alone was explained by the X2 and 14% by the variable X1, whilst total effect both X1 and X2 was 44% of total variation (Fig. 1C). In Nantong 1, only 2% of total variation in spider diversity was explained by the marginal effect of variable X1 (Fig. 1D).

RDA analysis (for testing the significance of each variance fraction) revealed strong effects of vegetation types (X1) and weighted PCNM (X2) on the abundance of different spider

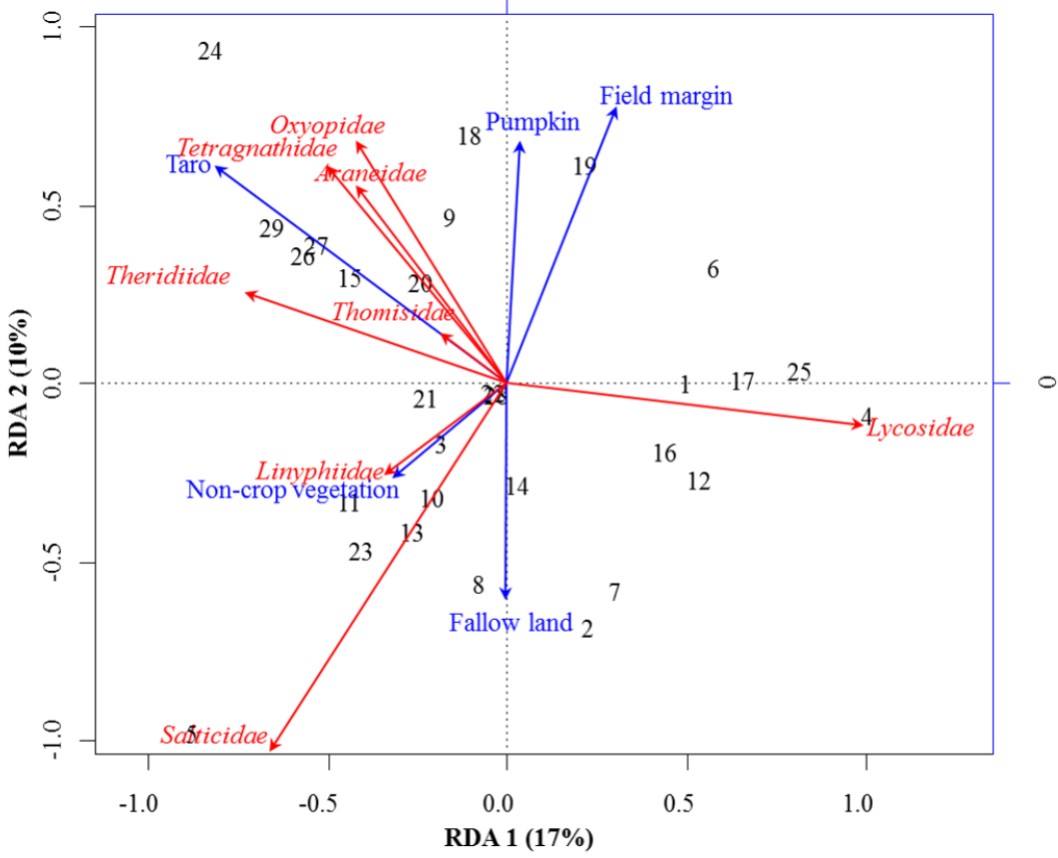

**Figure 2   RDA Triplot (RDA on a covariance matrix) of the spatial correlation between Hellinger transformed abundance of spider families and vegetation types surrounding the brassica field using PCNM as distance matrix.** The arrow length and direction correspond to the variance that can be explained by the environmental and response variables. The direction of an arrow indicates the extent to which the given factor is influenced by each RDA variable. The perpendicular distance between abundance of spider families and environmental variable axes in the plot reflects their correlations. The smaller the distance, the stronger the correlation. Numbers represents the sampling points in figure.

families in Minqing ($R^2 = 26\%$, adj $R^2 = 13\%$), but the overall significance of the model was lower ($p$-value $= 0.07$). Similarly, predictors X1 and X2 also showed strong effects for spider diversity ($H$) in Minqing ($R^2 = 19\%$, adj $R^2 = 14\%$) with lower significance of the overall model ($p$-value $= 0.28$). In Nantong 1, rda analysis showed strong effects of predictors X1 and X2 on the diversity ($H$) of spiders ($R^2 = 18\%$, adj $R^2 = 05\%$) with lower significance of the overall model ($p$-value $= 0.11$). Whilst, predictors X1 and X2 had less effects on diversity ($H$) of spiders in Nantong 2 ($R^2 = 10\%$, adj $R^2 = -03\%$) with very low significance of the global model ($p$-value $= 0.57$). RDA ordination showed that non-crop vegetation strongly supports the abundance of Linyphiidae and Salticidae at Minqing, while taro had particularly high in abundance of Araneidae, Oxyopidae, Tetragnathidae, Theridiidae and Thomisidae (Fig. 2). In Minqing, rda ordination for diversity ($H$) illustrated strong association of Thomisidae and Oxyopidae with non-crop vegetation, Salticidae and Lycosidae with fallow land, and taro, in contrast, had high diversity ($H$) of

Araneidae, Tetragnathidae and Theridiidae (Fig. 3A). However, in Nantong 1, non-crop vegetation held a greater diversity ($H$) of Araneidae (Fig. 3B), and Oxyopidae in Nantong 2 (Fig. 3C). Sweetpotato exhibited greater diversity of Tetragnathidae and Lycosidae at Nantong 1 (Fig. 3B), and Araneidae at Nantong 2 (Fig. 3C). Diversity of Oxyopidae showed strong positive association with Litchi in Nantong 1 (Fig. 3B). The field margins of brassica fields supported high diversity of Salticidae at Nantong 1 (Fig. 3B) and of Salticidae, Thomisidae and Lycosidae at Nantong 2 (Fig. 3C).

Community similarity/dissimilarity analyses between vegetation types, showed that brassicas share most of the spider families with other surrounding vegetation types in terms of abundance (Fig. 4A, Figs. S1A and S2A) and diversity ($H$) (Fig. 5A, Figs. S1B and S2B) (same colour in heatmap). The soil surface-associated hunting Lycosidae, however, showed strong differences in abundance and diversity ($H$) between different vegetation types in all experimental sites (Fig. 4A, Figs. S3A and S4A). Additionally, to assess the level of uncertainty in each cluster, the $p$-values (AU and BP) for each of the hierarchical clusters were calculated using bootstrap resampling techniques. Attributes of spider family abundance and diversity ($H$) are examined and hierarchical clustering performed. Values on the edges of the clustering are $p$-values (%). Red values are AU $p$-values and green values are BP $p$-values. Clusters with AU $p$-values >95% are significantly supported by the abundance (Fig. 4B, Figs. S1C and S2C) and diversity data of spiders (Fig. 5B, Figs. S1D and S2D). For example, abundance of spiders in Minqing (Fig. 4B), the cluster labelled 4 in Fig. 4B the observed AU $p$-values are 90%, 96%, 81% and 77%, whilst, observed BP values are 44%, 40%, 43%, and 37%, respectively, and the cluster dendrogram with 96% AU $p$-value were significantly supported by the spider abundance data.

Spatial autocorrelation patterns were found to be highly significant ($P < 0.001$) for the abundance of spiders in Minqing and Natong 2, and for diversity in Minqing and Nantong 1. The spatial weighting matrix maps, based on the xy-coordinates of each sampling point, associated with the dbMEM eigenfunctions for Minqing, Nantong 1 and Nantong 2 are shown in Fig. 6A, Figs. S3A and S4A, respectively. The significant spatial correlation model for Minqing, indicated that brassicas, non-crop vegetation, field margins, fallow land and taro were the vegetation types spatially associated with greater spider abundance (Fig. 6B) and diversity ($H$) (Fig. 6C). Similarly, for Nantong 2 brassica, field margin, sweetpotato and non-crop vegetation were spatially associated with greater spider abundance (Fig. S3B). Moreover, significant spatial autocorrelation was found only for spider diversity ($H$) in Nantong 1; where litchi, sweetpotato and non-crop vegetation exhibited strong positive spatial autocorrelation with the diversity ($H$) of spiders (Fig. S4B).

## DISCUSSION

Mixed cropping systems that include perennial crops, non-cropped and non-sprayed zones, offer a relatively stable environment, increasing the potential for alternative and source habitat for the conservation of natural enemies (*Blitzer et al., 2012*; *Marc & Canard, 1997*; *Rypstra et al., 1999*; *Schmidt & Tscharntke, 2005*). Among predator taxa that can be important are spiders that attack pests as diverse as *Spodoptera littoralis*

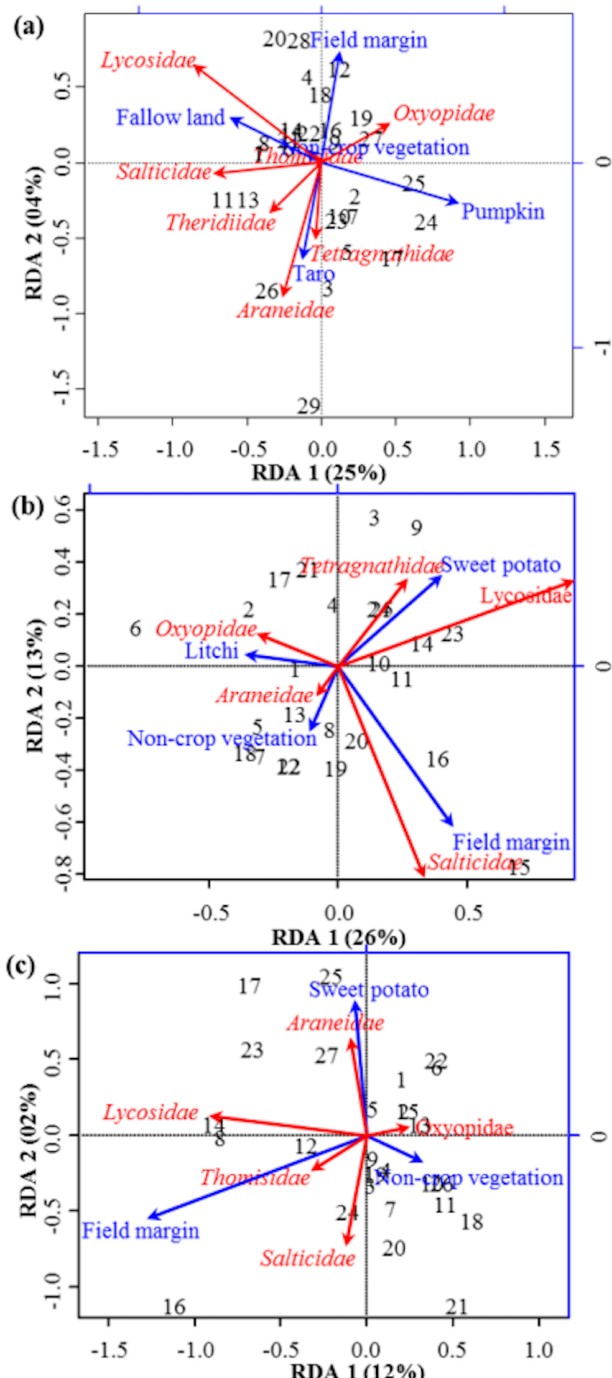

**Figure 3** **RDA Triplot (RDA on a covariance matrix) of the spatial correlation between Hellinger transformed diversity (*H*) of spider families and vegetation types surrounding the brassica field using PCNM as distance matrix (A) at Minqing, (B) at Nantong 1 and (C) at Nantong 2.** The arrow length and direction corresponds to the variance that can be explained by the environmental and response variables. The direction of an arrow indicates the extent to which the given factor is influenced by each RDA variable. The perpendicular distance between abundance of spider families and environmental variable axes in the plot reflects their correlations. The smaller the distance, the stronger the correlation. Numbers represents the sampling points in figure.

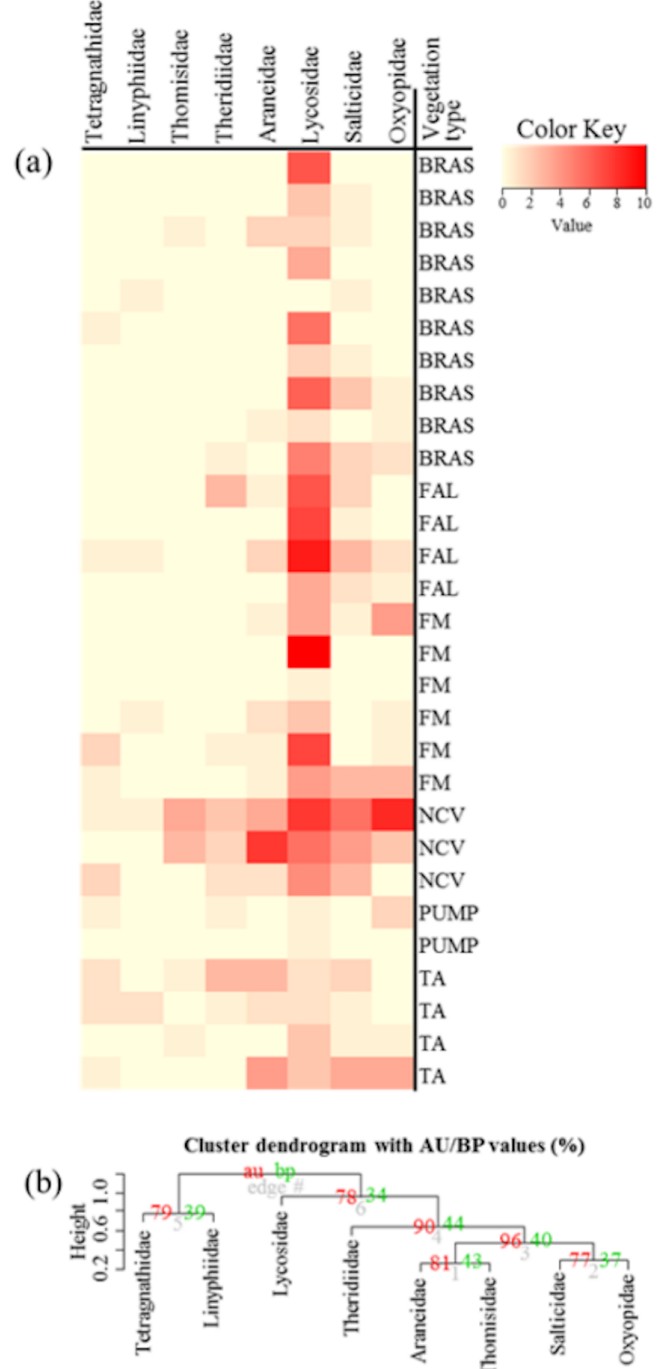

**Figure 4** (A) Heatmap based on hierarchical clustering using Bray–Curtis resemblance matrix of spider taxa abundance at Minqing, where: "BRAS", Brassica; "PUMP", pumpkin; "FAL", fallow land; "TA", taro; "NCV", Non-crop vegetation; and "FM", Field margin. (B) Cluster plot to test the goodness of hierarchical clustering for abundance of spider families at Minqing. Values at branches are approximately unbiased (AU) *p*-values (left), bootstrap probability (BP) values (right), and cluster labels (bottom). Clusters with AU > 95 are consider to be significant.

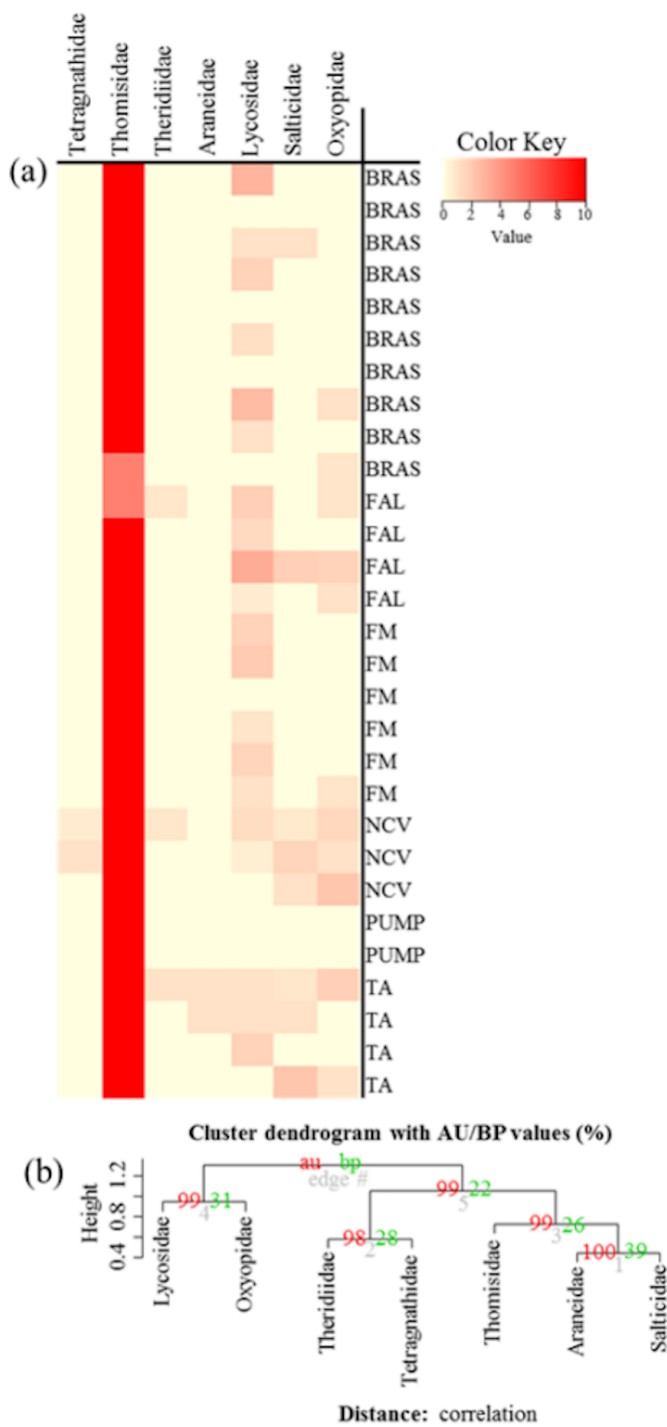

**Figure 5** (A) Heatmap based on hierarchical clustering using Bray–Curtis resemblance matrix of spider taxa Shannon diversity at Minqing, where: "BRAS", Brassica; "PUMP", pumpkin; "FAL", fallow land; "TA", taro; "NCV", Non-crop vegetation; and "FM", Field margin. (B) Cluster plot to test the goodness of hierarchical clustering for Shannon diversity of spider families at Minqing. Values at branches are approximately unbiased (AU) *p*-values (left), bootstrap probability (BP) values (right), and cluster labels (bottom). Clusters with AU > 95 are consider to be significant.

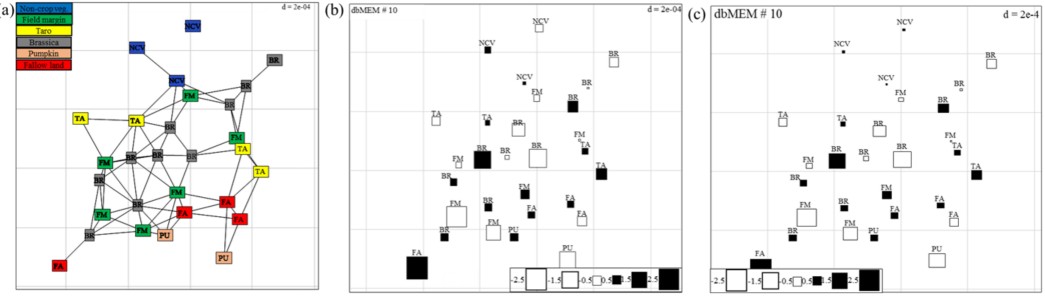

**Figure 6** (A) Map showing the 29 sampling points (~10 m apart) in Minqing computed using geographical sampling distance matrix. Bubble plot maps based on the forward selection to identify the significant dbMEM spatial model among all dbMEM eigenfunction models of spider's (B) abundance and (C) Shannon diversity; showing the relative importance of spider's abundance and diversity along with their spatial distribution. The size of the square box represents spider's abundance and diversity in each eigenvector, ranging from white (largest negative value) to black (largest positive value).

(*Mansour et al., 1980*) and aphids (*Birkhofer et al., 2008*). It is known that spider assemblages rather than individual, dominant species are important for pest suppression (*Riechert & Lawrence, 1997*; *Riechert & Bishop, 1990*) but—in contrast to non-spider taxa—we currently have a poor understanding of how to manage agroecosystems to best promote biological pest control by spiders. Moving beyond the generalization that non-crop vegetation can potentially suppressing pest populations by promoting functionally different groups of natural enemies (*Bianchi, Booij & Tscharntke, 2006*; *Boller, Häni & Poehling, 2004*; *Gurr et al., 2017*; *Thies & Tscharntke, 1999*) is a key challenge in applied ecology. Addressing this requires empirical evidence on the effects of differing vegetation types on associated abundance and impact in nearby focal crops but work of this type requires labour intensive surveys with associated laboratory sorting. Such field work can also can be stymied by unexpected events such as floods that lead to small sample sizes and data sets that are difficult to analyze with conventional statistical approaches. Our results suggest that more advanced statistical approaches offer the scope to deal with this dual challenge of ecology and data analysis.

In our study, spider community structure was clearly shown to vary among vegetation types. There was high variance observed for spider abundance among the different vegetation types at the scale of a few meters from the brassica crops in Minqing, whilst spider diversity ($H$) was mostly a function of spatial distance and its combined effect with adjacent crop and non-crop habitats. These results suggest the patchiness of spider distribution in brassica production systems and was much stronger for cursorial families (Lycosidae and Thomisidae) as compared with web-builders (Araneidae, Linyphiidae, Tetragnathidae), a finding that is broadly consistent with *Blitzer et al. (2012)* and *Schmidt et al. (2003)*. This may reflect differences among the vegetation types for bare ground would favour movement of cursorial spiders (ground-runners), unimpeded by vegetation structure. Whilst, vegetation type influenced spider abundance, diversity was less more strongly influenced by weighted PCNM matrix (distance between sites with special focus on neighbouring sites). This suggest that surrounding vegetation nearby the brassica field

affects the spider abundance at a local-scale (up to few meters from focal crop). This may relate to the structure and permanence of vegetation types, both of which affect the scope for a given vegetation type to provide alternative food or shelter resources and thereby drive the assemblage and diversity ($H$) of spiders (*Langellotto & Denno, 2004*; *Schmidt & Tscharntke, 2005*; *Thies & Tscharntke, 1999*).

Whilst some spider species tend to dominate predator communities in crop fields and are considered as "agrobionts" (*Samu & Szinetár, 2002*), it is not clear if these species generally prefer crop fields over other vegetation types and to what degree they may discriminate between crop types. Specifically, in brassica agroecosystems with high levels of disturbance from insecticide use, planting, and harvest events, adjacent crop and non-crop vegetation can play a vital role in the local conservation of spiders. Our results illustrate that, for most of the spider families, abundance is strongly associated with perennial or dense, bushy vegetation types (taro, non-crop vegetation and pumpkin) nearby the brassica fields. This finding is in accordance with (*Schmidt et al. (2003)* and *Schmidt & Tscharntke (2005)* that adjacent perennial vegetation can strongly influence the abundance and diversity of natural enemies. This may be because these vegetation types offer a refuge from disturbance and in which alternative food sources are present (*Halley, Thomas & Jepson, 1996*; *Topping, 1999*; *Topping & Sunderland, 1994*). In contrast to abundance, patterns of spider diversity ($H$) in our study demonstrate strong association of non-web building spiders (Lycosidae, Salticidae, Thomisidae and Oxyopidae) with fallow land and brassica fields (e.g., *Carvalho & Cardoso, 2014*; *Uetz, Halaj & Cady, 1999*). This may be a consequence of their mode of hunting, since such habitats have relatively large areas of bare ground for dispersal and foraging (*Schmidt & Tscharntke, 2005*). For web building families (Theridiidae, Araneidae, Tetragnathidae and Linyphiidae), diversity showed a strong association with the taro, sweetpotato and non-crop vegetation, which may be due to the availability of more relatively complex plant structures for building webs, potentially complemented by the low disturbance regime of the fallow land (*Schmidt & Tscharntke, 2005*; *Thies & Tscharntke, 1999*; *Topping, 1999*). Overall, these results suggested different habitat requirement for these two functional groups of spiders, further driving resource differentiation. Distinct preferences, in terms of niche requirements for particular habitat—composed of certain plant diversity—are known for spiders, (e.g., *Bonte, Baert & Maelfait, 2002*; *Griffin et al., 2008*). Such preferences offer scope for manipulative use to promote the ecosystem services of biological control by spider functional groups that are the able to partition the prey resource and achieve high levels of suppression. These results provide a foundation for future research to further unravel the underlying mechanisms for the patterns observed here; for example, distribution and assemblage of spider species caused as a result of plant structural diversity in various cover types or caused by various agronomic practices and the role of broader landscape in aerial dispersal of spiders.

In terms of advancing analytical approaches for handing data sets of the type dealt with here, hierarchical clustering is shown to be a useful for measuring community dissimilarities. In this study, we move beyond the measuring of diversity within the sites and we investigated the $\beta$-diversity by assessing similarity of the spider assemblages among the sampled habitats (*Aanderud et al., 2015*; *Warnes et al., 2016*). Results of $\beta$-diversity

analysis showed commonality in most of the spider taxa abundance and diversity between brassica and adjacent crop and non-crop vegetation types. This suggests that certain adjacent crops (taro, sweetpotato and pumpkin) and non-crop habitats (non-crop vegetation and field margins) shared spider taxa with brassica fields, so these may provide especially useful refuges and serve as donor habitat for spiders spilling over into brassica crops following a disturbance event such as replanting, insecticide use or flood.

The statistical approaches used in the present study show utility for extracting, from data sets of modest size, testable hypotheses that can explore underlying mechanistic phenomena related to spill-over patterns and confirm the relative importance of difference vegetation types as source habitat for a given focal crop type. It is becoming necessary that ecologists incorporate spatial autocorrelation patterns into ecological models, and the analysis of population dynamics, and species distribution (*Blanchet, Legendre & Borcard, 2008*). Our results detected significant spatial autocorrelation patterns between the numbers of spider individuals at different sampling points, and revealed highly significant spatial correlations between the abundance of the spiders with field margins, taro, non-crop vegetation and sweetpotato. The spatial eigenvectors method proved to be sensitive for detecting spatial patterns in the present data despite it being constrained by natural factors. Accordingly, our study also expands the methodological foundation for agroecological studies of ecosystem providers for future research.

During the last few decades, the loss of biodiversity and ecosystem function in modern agroecosystems has been a major and growing concern of agroecological researchers (*Bommarco, Kleijn & Potts, 2013*; *Millennium Ecoysystem Assessment, 2005*; *IPES-Food, 2016*; *Potts et al., 2016*). Our study illustrates the importance of non-crop plants nearby to crop fields to promote conservation biological control strategies for spiders and generates testable hypotheses for future studies. For example, there is a need to measure and track actual rates of spider movement between the habitat types used in the present study, in order to determine if the predicted habitat types really are key donors of spider colonization and recolonization for brassica crops. In addition, patterns of spider movement need to be studied in relation to disturbance events. More generally, future research should extend to testing the temporal effects of farm management practices (i.e., cropping patterns, chemical inputs) interacting with agricultural landscapes heterogeneity (compositional and configurational) on organizational and functional levels of agroecosystem. These are the major factors which drive the distribution, structure and composition of spider community in agroecosystems.

## ACKNOWLEDGEMENTS

We thank Dr. David J. Perovic for advice on data analysis; Professors Guang Yang and Weyi He (Institute of Applied Ecology, FAFU, China) for advice; Saif-ul-Islam (College of Plant Protection, FAFU, China), Han Liwei and Zhang Hanfang (Institute of Applied Ecology, FAFU, China) for technical support; and Mrs AC Johnson (Charles Sturt University) for manuscript editing.

### Funding

This study was financially supported by a Chinese Government Thousand Talents fellowship to Geoff M. Gurr. The funders had no role in study design, data collection and analysis, decision to publish, or preparation of the manuscript.

### Grant Disclosures

The following grant information was disclosed by the authors:
Chinese Government Thousand Talents.

### Competing Interests

Geoff M. Gurr is an Academic Editor for PeerJ.

### Author Contributions

- Hafiz Sohaib Ahmed Saqib conceived and designed the experiments, performed the experiments, analyzed the data, wrote the paper, prepared figures and/or tables, reviewed drafts of the paper.
- Minsheng You and Geoff M. Gurr conceived and designed the experiments, wrote the paper, reviewed drafts of the paper.

### Data Deposition

The R-codes and data have been uploaded as Supplemental Files.

### Supplemental Information

Supplemental information for this article can be found online at http://dx.doi.org/10.7717/peerj.3795#supplemental-information.

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
