# Peer review of "Multivariate ordination identifies vegetation types associated with spider conservation in brassica crops"

_PeerJ, doi:10.7717/peerj.3795_

## Round 0.1 · original submission · Major Revisions

The authors present an interesting research that confirms for Conservation biological control the importance of vegetation other than the focal crop. Biodiversity and environmental heterogeneity it is of great importance to maintain the populations of the natural enemies. However, it is very difficult to be ecologically optimized. Thus, the authors attempt to measure it using advanced experimental design and statistical analyses. This is the strength and in turn to be the weakness of their contribution. Reviewers require more effort by authors to further explain the manuscript methodology section (experimental design and statistical analyses) that unfortunately limits the discernment of the work done.

So I must ask the authors please to follow the suggestions of the reviewers carefully to improve the solidity of the new knowledge presented by the common good, of improving the science of conservative biological control, upon before manuscript definitive acceptance

It is attached the comments of the reviewers and an article with comments.

Reviewer 1 ·

Basic reporting

Clear, unambigous, professional English language used throughout.
Intro and background were used to show content.
Structure conforms to Peerj standards, discipline norm, or improved for clarity.
Figures are relevant, high quality, well labelled and described.
Raw data supplied.

Experimental design

Original primary research within Scope of the journal.
Research question well defined, relevant and meaningful. It is stated how the research fills an identified knowledge gap.
Rigorous investigation performed to a high technical and ethical standard.
Methods described with sufficient detail and information to replicate.

Validity of the findings

Data is robust, statistically sound and controlled.
Conclusions are well stated, linked to original research question and limited to supporting results.
Speculation is welcome, but should be identified as such.

Additional comments

The authors present an interesting research that confirms the importance of environmental heterogeneity when establishing populations of generalist predators useful for agricultural crops. In addition, the manuscript presents statistical tools that can be useful in research on agricultural crops that are conditioned by seasonality and extreme weather phenomena. If there is a weakness, it is in the way that some of the results are presented (as I have noted above) which should be improved upon before Acceptance.

Various references to supplementary material are made in the results section. It becomes difficult to follow the text with so many references to different figures that do not appear in the main manuscript that are mentioned profusely in the results. Please clarify it.

Specific comments:
Line 71: The reference to spiders studied as belonging to 'Araneida' is risky because the taxonomic organization of spiders (in terms of order and suborder) is still discussed. In any case if the authors wish to maintain their denomination they should explain which criterion they follow to define the spiders like 'Araneida' and not 'Araneae'.
Line 108: Indicate the bibliographic sources used to identify the spiders.

Reviewer 2 ·

Basic reporting

English should be carefully checked throughout. I pointed out a few issues and some paragraphs that need re-writing.

More background literature should be added to hypotheses and discussion. The discussion feels incomplete. With all the analyses done a lot more can be discussed.

Experimental design

The methods have some issues pointed out in the PDF. Also, the methods are not clearly described and make it confusing to understand.

Validity of the findings

The data would me more robust by adding suggested analyses (see PDF).

Discussion and conclusion need to be re-written and or expanded.

Additional comments

This is a great and valid contribution, however the paper has several flaws all pointed out in the PDF. Mainly the written english, some of the methods and the discussion.

Annotated reviews are not available for download in order to protect the identity of reviewers who chose to remain anonymous.

Reviewer 3 ·

Basic reporting

The manuscript is globally understandable. However, there are numerous grammatical and spelling mistakes, as well as syntax errors that need to be corrected.

The introduction and particularly the discussion are often too general and miss concrete examples, especially the discussions about the pests/natural enemies/vegetation types. In addition, some parts of the discussion (quite general) would best fit in the introduction.

Some information is missing in the figure captions. Figure 6 is illegible. Some figures included in the Supplemental material would be best in the MS (e.g., RDA graphs). Also, one table could be added to summarize some pertinent information about the methodology.

Main results needed to answer the research question are provided. However, numerous figures are part of the Supplemental material (only figure about site 1 is provided in the manuscript).

Experimental design

The manuscript entitled "Multivariate ordination identifies vegetation types associated with spider conservation in brassica crops" meets the aims and scope of the PeerJ journal.

The purpose of the study is well stated and relevant, and the justification of the research is provided. However, the hypothesis is too general and could to be more precise. It would be interesting to make hypotheses about which habitat types would provide the highest abundance and diversity of spiders in brassica fields and why.

The sampling design is acceptable to answer the research question. However, a more rigorous work could have been done by identifying the spiders to the species level (instead of the morphospecies) or by conducting the experiment over several years.

The most important issue with this manuscript is the methodology section (experimental design and statistical analyses) that needs to be further explained. There is a lack of details at several places in the manuscripts that limits the comprehension of the work done. For instance, a figure of the sampling design or a Table that summarizes all the essential information may help to understand how the samplings was performed (e.g., number of sampling points per site, number of sampling points per habitat types and per site). The statistical analyses section also needs further details to completly understand what was done and why.

Validity of the findings

The authors succeed to identify some habitat types that seem important for the conservation of spider abundance and diversity. However, the results greatly differ between the 3 sites and discussion remains often too general. The discussion may be improved by being more specific and, for instance, by giving specific recommendations (overall, which habitat types exactly should be present near brassica crops to provide a high abundance and diversity of spiders?).

The data consisted in only one year of sampling and, due to an exceptional event (typhoon), not all the sampling dates were taken into account in the analyses. However, the data seem correctly analyzed.

Because of the numerous analyses done, the "take-home message" is not always clear and needs to be better identified. The discussion may also be improved by adding critics of the methodology/results found. For instance, results and discussion should be moderated since 1) the study was performed in only one year and that an exceptional event happened during this year (typhoon), and 2) identification of the spiders were not performed at the species level (but morphospecies, so with possible bias).

Additional comments

General comments
The manuscript is globally understandable; however there are numerous grammatical and spelling mistakes, as well as syntax errors that need to be corrected.
The major issue of this article is the lack of important information, especially in the methodology and result sections, which limits the comprehension of the experimental design, the statistical analyses performed and the results obtained.
In addition, the introduction and particularly the discussion are often too general and miss concrete examples. The discussion could also be improved by adding critics of the methodology used and results found, and comparisons with other articles observing effects of habitats type on spider abundance and diversity (not only natural enemies in general).

Abstract
L16: “the importance of vegetation…”: You may precise “Natural vegetation”.
L17: Replace “disturbance” by “disturbances”
L18: Add “the” before “types of vegetation”
L19: Remove “into”
L19-20: “Here we explore… and non-crop vegetation”. You may consider rewriting this sentence, it sounds odd.

Keywords
I am not convinced about the term “Ecological Engineering”

Introduction
General comments
The introduction is overall well written but often lacks of details. It is often too general and may be improved by given some precise examples (see specific comments below).

Specific comments
L38-45: This part could be improved by adding an idea of the period of time you refer to. “Anthropogenic activities” since when (from few years ago, decades ago…)?

L71-74: More information should be given regarding the spider communities found in brassica crops. You mentioned it in the discussion but it can also be described in the introduction. Actually, some information given in the discussion should actually be part of the introduction (e.g. L236-241). It is also important to precise in the introduction (not only in the discussion) that spiders are generalist predators. Furthermore, essential information is missing: Which brassica pests can be controlled by spider in brassica fields? Please provide more details.

L71-84: The last paragraph could be divided into 2 paragraphs: 1) One paragraph about spider communities found in brassica crops and their importance to control pests; so you can add more details about their biology and ecology (e.g., do they overwintered in non-crop habitats adjacent to crop fields?) and 2) second paragraph about the purpose of the study and hypotheses.

L81-84: The hypothesis remains very general. “different vegetation communities would share the spider fauna of brassica fields to varying degrees…”. Did you have an idea of which habitat types would share more species with brassica crops?

Material and methods
General comments
The most important issue of this manuscript is the lack of important information in the methodology. A lot of details are missing and prevent the complete comprehension of the study.
The methodology section could be improved by adding a table that compiled all relevant information for the comprehension of the sampling design: sites, total number of sampling points per site, number of each vegetation type sampled per site, details about the sampling period (exact dates of sampling)... Also, a figure of the sampling design may help understanding how the sampling where performed (e.g., how many sampling points in each habitat type?).
Finally, I wonder whether one year of data is relevant enough to make conclusions, as effects of landscape on insect communities can greatly vary from one year to another, and considering that one event had perturbed the samplings.

Specific comments
L90: Which brassica species? Were there different species? If so, it would be important to list the different species.

L91: “50 x 50 m grid”. Why this distance? You need to justify the choice of this specific scale. Is that because of the dispersal ability of spiders?

L95: “not uncommon”. So it is common?

L98-99: First, it is not clear in the MS whether you have 25-29 sampling points for each one of the 5 sampling periods or for the entire sampling period. Also, when looking at your dataset, it is not clear how many sampling grids you had per site. Did you initially have 30 sampling points for each date and site? In Minquing for instance, in the data file, the sampling points are numbered from 1 to 30, but 19 is missing. Why? So there are a total of 29 points, but only 28 in the other data file (Data_set_Sp). How can you explain that? All these details are essential to understand the experimental design. If there were 30 sampling points at the beginning you should say so. Same problems appeared with the other sites (Nantong 1: 25 or 26 grid points? Nantong 2: 27 or 28 grid points?)
Second, how many samplings were performed in each vegetation types? And how many vegetation types did you consider? These details could be summarized in a Table.

L99: “all vegetation types”: I do not understand the classification of the different vegetation types. This must be clarified in the methodology section. What does “non-crop vegetation” refer to? It is mentioned in the methodology (L92-94) that “non-crop vegetation consisted of adjacent field margins and fallow fields… as well as uncultivated areas…”. But in the results, non-crop vegetation, fallow and field margin are separated. So I wonder what were included in the “non-crop vegetation” category.

L100: “five occasions”. You need to precise the different sampling dates (could be included in a Table, as suggested before). Was all the vegetation types sampled during the 5 sampling dates? It is not clear.

L103-104: So how many sampling dates do you have overall?

L108-109 “(genus in some cases) and assigned to morphospecies”. It is not clear whether the analyses take into account the genus or morphospecies level. You need to be more precise. I assume that the morphospecies was used to calculate the diversity index. If so, it should be mentioned. You don’t need to say that some species were identified to the genus level if you don’t do analyses at this level.

Statistical analyses
General comments
The statistical analyses seem ok to me there are a lack of details and missing important information at numerous places, which limits the comprehension of the analyses done. I am familiar with RDA and variation partitioning analyses, but less with cluster analyses so I gave fewer comments on this part. In addition, the authors need to verify that the correct references are cited for all R packages mentioned.

L114 “spider abundance and diversity data”. What was the unit of replication in the analyses? Did you use the total abundance or mean abundance? As I understand when looking at your dataset is that you keep the data per sampling point. But, did you pool all the data from the 5 sampling periods? Did you use the genus or morphospecies level to calculate the Shannon index of diversity? All of this is not clear and need to be mentioned.

L117: Hellinger transformation: Also used when a lot of zeros in the matrix, like in your dataset.

L121: “differing vegetation types”. “different” instead of “differing”. What vegetation types did you include in the variation partitioning?

L121 “principal” instead of “principle”

L121-122. You may better explain why you decide to perform a variation partitioning between the vegetation type and the principal coordinates of neighbor matrices (PCNM). You may better explain what the PCNM is about.
L124: “effects of distance”: the distance between what and what? Is that the distance between the different sampling points within each site?

L128: It is said that the “adjacent land cover” is different between sites. It would be good to have a more precise idea of these differences. Can you add a characterization of the 3 different sites? For instance in a Table (e.g., including the proportion of the different land covers)?

L129-140. I am less familiar with this sort of analyses, so I am less able to criticize thoroughly this part. However, I have one comment about the dissimilarity index.

L145: “occurrence of spiders”. Occurrence per site? Per sampling point? For the entire season or per sampling dates?

L147: “spatial correlation”. Did you mean “Spatial autocorrelation”?

L148-156. The term “spatial variation map” is not clear. In addition, this part looks more like results than methodology.

L150-151. I am familiar with multivariate analyses and forward selection but I don’t understand what you did here. Why the “spatial variation maps” were included in a forward selection.

L155; “Whist”. ? Verify English.

L159: “p-value<0.00”. Replace by “p-value < 0.001”).

L159-161: You need to cite the references for all the R packages used.

Results
L163-164: It is important also to have the details of the captures per site (abundance and diversity). Moreover, why didn’t you identify to the species level? Did you confirm the different morphospecies with an expert? If not, you should discuss about it in the discussion (you may consider having an over/under-estimation of the diversity).

L164-171: “Variance partitioning showed that abundance of spider families was vegetation-type dependant…”. Important information is missing here. First, Fig 1a misses the total value of the effect of the variable X1=Vegetation and X2 = PCNM (by the way, you must change the second X1 in Figure 1 to X2, because you have 2 variables called X1). The value 0.13 means that the vegetation type alone explains 13% of the variation in spider abundance, which is not very high. What is the total effect of the variables X1 and X2 (unique contribution of each variable + share effect)? If the residual effect is 0.94, than the total effect of X1 and X2 is 6%, which is quite low.
Second, you precise in the following sentence that you have tested “the significance of each variance fraction”, but you precise only one p-value (p=0.07). Is that the p-value for the global model (including effect of X1 and X2)? Also, you mentioned that there are “strong effects” but I disagree with that. All the p-values you mentioned are > 0.05. I disagree also with the term “strong effect”. Actually, you have a strong effect when the R2 of your model has a high value, and the model is highly significant when p-value is very low. The description of the results needs to be rewritten.
Third, you mentioned only the results of the variation partitioning for spider abundance in Minqing, and spider diversity in Nantong 1 and Minquing. What are the results of the other variation partitioning? Even if you found no effect, you have to say it. It is not clear.

L174-176: I am not totally in agreement with you observations. According to me, when looking to Fig 3, Non-crop vegetation are associated with Thomisidae (maybe Oxyopidae), but Salticidae and Lycosidae are more associated with Fallow land than Non-crop vegetation.

L177 “pumpkin showed a strong association with Oxyopidae”. I disagree. The association is not so strong.

L177-183 It would be interesting to have all the RDA graphs in the MS.

L180: “Oxyopidae showed …” Oxyopidae abundance or diversity? Please, be consistent in the structure of the sentence. Put the species family first (e.g., Oxyopidae showed strong association with….) or the habitat first (e.g., Non crop vegetation are associated with…)

L181: Replace “Fig 2a” by “Fig S2a”

L184-185: Quite general. “most of the spider families” “Other surroundings vegetation types”. Can you be more precise. It is difficult to understand the results for people that have never done such analyses. Please explain in more details what do you mean by “same coulour in heatmap”. You should explain more precisely, at least in the figure caption, what does the color key correspond to (what is the unit of the value 0-10?).

L186-187: “Lycosidae showed strong differences in abundance between…”. You can add “and diversity” so that you don’t have to repeat Lycosidae in the next sentence.

L188-189: “Oxyopidae depicted strong differences in diversity among different vegetation types…(different colour)”. I disagree for Nantong 1 and Nantong 2 because it is actually the same colour for all the vegetation types (Fig S3b and S4b).

L189: Remove one “Fig”

L192-193: What does it mean concretely to have such AU-value and BP values? Can you explain briefly? It is important to give some precisions if you want people that are not familiar with such analyses to understand your results. Here, it looks like only a series of values, without explanations.

L196: “Spatial autocorrelation”: between sampling points?

L197: Spatial correlation or autocorrelation? Do you refer to the same thing? Please explain what you mean. Be consistent in the writing and correct when needed

Figure 1: What is the meaning of the intercept?
Figure 2: Which site?
Figure 4-5: Did you sampled in brassica fields? It is not clear n the methodology. Why Brassica appears in Figures 4 and 5 and not in the RDA graphs? In the figure caption,
Figure 6 is very small and illegible.

Discussion
General comments
First, several part of the discussion are too general and need some precisions. Some hypotheses could also be added to explain some of the results found.
The discussion may also be improved by adding critics of the study (e.g., over/underestimation of the spider diversity because no identification to the species level…). What could have been done or analyzed (e.g., analyze of the effect of the vegetation type on the distribution of spider depending on the sampling periods…). Results would have been different if more than one year studied?

Specific comments
L208: “and non-cropped and non-sprayed zones”: You may rephrase (the two “and” are a bit redundant).

L210: “various pests”: can you give examples of pests controlled by spiders?

L214-216: “Requires….to require”: Replace one of the two “require”

L218-220: What did you mean by “conventional statistical approaches” and “advanced statistical approaches”? I am not sure that the pertinence of this section.

L224: 2a function of spatial distance”: Distance between what and what?
L224: “its intercept”: What does it mean exactly in an ecology point of view (not only statistics)?

L225-228: So it means that “cursorial families” are more widely distributed than “web-builders”? Can you explain why or at least give some hypotheses?

L227-228: “as observed (Blitzer et al. 2012…”. Correct the end of the sentence: “as observed by Blitzer et al. (2012) and Schmidt et al. (2003).

L230: “spatial distance”. It is not clear (between what and what?).

L230: “cabbage”. In the rest of the MS, the word “brassica” was used. Please, be consistent in the terminology.

L231: “meso-scale”. Is 50m really a meso-scale for spiders? I would have used the qualification “local scale” instead. Unless you can have justifications.

L232: “alternative food”: Can you be more precise (e.g., examples of preys)?

L236-214: This part should be part of the introduction.

241-244: I do not understand. Are you referring to your results or to other authors (Schmidt et al 2003…)?

L244-246: Too general. “alternative food”, “place where spider have time and space…” Can you be more precise? What kind of food and what kind of habitat?

L252: Replace “this” by “which”

L253-255: “may be because of the availability…”. You may replace by “may be due to the availability”. Remove “may also be”.

L267 “vary in space”: Did you mean between habitats?

L269: “certain adjacent crop”: Can you be more precise and give some examples? Can you give some recommendations? Which habitat type should be present near brassica crops to provide a high abundance and diversity of spiders?

L278-280: Spatial correlation or autocorrelation?

L280: Replace “with” by “and”. “various vegetation types”: Can you precise?

L282: I don’t see the need here to emphasis again about the typhoon.

L283-284: I am not totally convinced.

L286: Replace “is” by “has been”

L287: Replace “likely” by “great”

L288: Replace “nearby to crop fields” by “nearby crop fields”. Replace “conservational” by “conservation”

L293: Replace “Further” by another word (In addition, furthermore…)

L294 “temporal effect”. As you have several sampling periods, it would have been very interesting whether the effect of vegetation type on the distribution of spider differ over time.

---

## Round 0.2 · accepted · Accept

I considered that the authors had provided a broad and sufficient response to the reviewers' requests.